# OpenReview forum: "Check Yourself Before You Wreck Yourself: Selectively Quitting Improves LLM Agent Safety"
_NeurIPS.cc/2025/Workshop/Reliable_ML — NeurIPS 2025 - Reliable ML Workshop_

### Official Review · Reviewer_2ddJ · 2025-09-14
**Interesting and useful idea, but need more insights**

**Rating:** 5
**Confidence:** 3

**Review:**

*Summary*

This paper discusses how we may add quitting options and instructions to LLM agents to exit certain harmful actions. Precisely, this is subdivided into options to quit (simple quit) vs explicit instructions to quit (specified quit). In the experiments this work demonstrated that both the quit options improved the safety score, with specified quit showing a better improvement compared to simple quit. This gives the intuition that LLM agents have the bias to complete their actions even if quitting is an option. Meanwhile, the helpfulness score was minimally impacted. The experiments also show that the proprietiary model generally shows a higher "quit" rate and therefore an increased safety score. The paper also justifies the "quitting" option by showing a positive correlation between quit rate and safety score, suggesting that the agents can detect the "harmful" behaviors users specified fairly well.

*Strengths*

This paper demonstrates an interesting idea whereby one can add some explicit quitting instruction to LLM prompts to halt harmful behaviors. This is substantiated by experimenting on a few well-known LLMs.

*Weakness*

Here are some suggestions to the authors on what to add to improve the paper:

- Insights: why is it the case where the proprietary models have higher quit rate (and better in safety improvement) compared to the open source ones? A few examples connecting the differences between proprietary and open source ones (e.g. training protocol or prompts fed during pretraining) would help.
- More experimentation: the paper considers simple vs specified quit, but not a spectrum of "specified". E.g. in the specified quit setting, does it matter if we only list 1 example scenario where the agent must quit (and leave the rest to the agent's discretion) or all such scenarios? How about the level of details of such prompts? (E.g. "If you cannot rule out negative consequences" vs "If you cannot rule out negative consequences, such as (a), (b), (c), (d), etc).
- Clarity: how to interpret the safety or helpfulness score from 0 to 3? From what I read it seems like I am supposed to treat it as a black box tool from ToolEmu (other than comparing the value, i.e. which one is higher).

Overall, I'd like to see this work expanded into something where we can understand how to design prompts to make agents quit safely, instead of seeing examples of how they work.

---

### Official Review · Reviewer_9j4d · 2025-09-19

**Rating:** 6
**Confidence:** 4

**Review:**

Summary

The authors propose quitting as a safety mechanism for LLM agents operating in multi-turn, high-stakes environments. Instead of forcing completion, agents are instructed to explicitly quit when uncertain or when potential risks are detected. Using the ToolEmu benchmark with 144 scenarios and 12 models, they show that specified quitting improves safety scores by 46 percent on average while reducing helpfulness by only 1 percent. Proprietary models respond more strongly to quitting prompts than open-weight models.

Strengths
1. This is a simple and practical intervention that directly addresses compounding risks in agent settings.
2. The evaluation is systematic, covering many models and scenarios.
3. Results demonstrate a favorable safety-helpfulness tradeoff, with quitting reducing unsafe actions at low cost. The work provides clear evidence that model prompts alone can substantially shift behavior.

Weaknesses / Limitations
1. The approach is a bit coarse: quitting halts tasks but does not explore intermediate strategies such as asking clarifying questions.
2. The method depends heavily on the ToolEmu benchmark, so generalization to real-world deployments is not guaranteed.
3. Results also show open-weight models benefit less, raising concerns about broader applicability.
4. Reasoning-oriented models like Claude or GPT-4o still defaulted to unsafe action unless quitting was explicitly emphasized, suggesting that reasoning capacity alone does not translate into safety without explicit directives.

Suggestions for Authors
1. Explore fine-tuning approaches to embed quitting behavior beyond prompting.
2. Consider deeper evaluation of reasoning models: can structured reasoning itself enable safe abstention without explicit quitting instructions?

Ethics
- The work is well-motivated by safety in high-stakes applications, and quitting as a safeguard is low risk.